# Co-expression analysis reveals interpretable gene modules controlled by *trans*-acting genetic variants

**Liis Kolberg\*, Nurlan Kerimov, Hedi Peterson, Kaur Alasoo\***

Institute of Computer Science, University of Tartu, Tartu, Estonia

**Abstract** Understanding the causal processes that contribute to disease onset and progression is essential for developing novel therapies. Although *trans*-acting expression quantitative trait loci (*trans*-eQTLs) can directly reveal cellular processes modulated by disease variants, detecting *trans*-eQTLs remains challenging due to their small effect sizes. Here, we analysed gene expression and genotype data from six blood cell types from 226 to 710 individuals. We used co-expression modules inferred from gene expression data with five methods as traits in *trans*-eQTL analysis to limit multiple testing and improve interpretability. In addition to replicating three established associations, we discovered a novel *trans*-eQTL near *SLC39A8* regulating a module of metallothionein genes in LPS-stimulated monocytes. Interestingly, this effect was mediated by a transient *cis*-eQTL present only in early LPS response and lost before the *trans* effect appeared. Our analyses highlight how co-expression combined with functional enrichment analysis improves the identification and prioritisation of *trans*-eQTLs when applied to emerging cell-type-specific datasets.

**\*For correspondence:**
liis.kolberg@ut.ee (LK);
kaur.alasoo@ut.ee (KA)

**Competing interests:** The authors declare that no competing interests exist.

## Introduction

Genome-wide association studies have been remarkably successful at identifying genetic variants associated with complex traits and diseases. To enable pharmacological and other interventions on these diseases, linking associated variants to causal intermediate phenotypes and processes is needed. A canonical example is the causal role of circulating LDL cholesterol in cardiovascular disease (*Ference et al., 2012*). However, discovering clinically relevant intermediate phenotypes has so far remained challenging for most complex diseases. At the molecular level, *cis*-acting gene expression quantitative trait loci (*cis*-eQTLs) can be used to identify putative causal genes at disease-associated loci, but due to widespread co-expression between neighbouring genes (*Wainberg et al., 2019*) and poor understanding of gene function, these approaches often identify multiple candidates whose functional relevance for the disease is unclear.

A promising approach to overcome the limitations of *cis*-eQTLs is *trans*-eQTL analysis linking disease-associated variants via signalling pathways and cellular processes (*trans*-acting factors) to multiple target genes. Although *trans*-eQTLs are widespread (*Võsa et al., 2018*), most transcriptomic studies in various cell types and tissues are still underpowered to detect them (*Aguet et al., 2019*). This is due to limited sample sizes of current eQTL studies, small effect sizes of *trans*-eQTLs, and the large number of tests performed (>$10^6$ independent variants with >$10^4$ genes). To reduce the number of tested phenotypes, co-expression analysis methods are sometimes used to aggregate individual genes to co-expressed modules capturing signalling pathways and cellular processes (*Stein-O'Brien et al., 2018*). Such approaches have been successful in identifying *trans*-eQTLs in yeast (*Parts et al., 2011*) as well as various human tissues (*Hore et al., 2016*; *Mao et al., 2019*; *Nath et al., 2017*) and purified immune cells (*Ramdhani et al., 2020*; *Rotival et al., 2011*). An added benefit of co-expression modules is that they can often be directly interpreted as signatures

of higher level cellular phenotypes, such as activation of specific signalling pathways or transcription factors (*Parts et al., 2011*; *Way et al., 2020*).

Gene co-expression modules can be detected with various methods. Top-down matrix factorisation approaches such as independent component analysis (ICA) (*Hyvärinen and Oja, 2000*), sparse decomposition of arrays (SDA) (*Hore et al., 2016*) and probabilistic estimation of expression residuals (PEER) (*Stegle et al., 2012*) seek to identify latent factors that explain large proportion of variance in the dataset. In these models, a single gene can contribute to multiple latent factors with different weights. In contrast, bottom-up gene expression clustering methods such as weighted gene co-expression network analysis (WGCNA) (*Langfelder and Horvath, 2008*) seek to identify non-overlapping groups of genes with highly correlated expression values. Recently, both matrix factorisation and co-expression clustering methods have been further extended to incorporate prior information about biological pathways and gene sets, resulting in pathway-level information extractor (PLIER) (*Mao et al., 2019*) and funcExplorer (*Kolberg et al., 2018*), respectively. Out of these methods, ICA, WGCNA, SDA and PLIER have previously been used to find *trans*-eQTLs for modules of co-expressed genes (*Hore et al., 2016*; *Mao et al., 2019*; *Nath et al., 2017*; *Ramdhani et al., 2020*; *Rotival et al., 2011*), but only a single method at a time. However, since different methods solve distinct optimisation problems, they can detect complementary sets of co-expression modules (*Stein-O'Brien et al., 2018*), with recent benchmarks demonstrating that there is no single best co-expression analysis method (*Way et al., 2020*). Thus, applying multiple co-expression methods to the same dataset can aid *trans*-eQTL detection by identifying complementary sets of co-expression modules capturing a wider range of biological processes (*Way et al., 2020*).

Another aspect that can influence co-expression module detection is how the data is partitioned prior to analysis (*Stein-O'Brien et al., 2018*). This is particularly relevant when data from multiple cell types or conditions is analysed together. When co-expression analysis is performed across multiple cell types or conditions, then the majority of detected gene co-expression modules are guided by differential expression between cell types (*Quach et al., 2016*; *van Dam et al., 2018*). As a result, cell-type-specific co-expression modules can be missed due to weak correlation in other cell types (*van Dam et al., 2018*). One strategy to recover such modules is to perform co-expression analysis in each cell type separately (*Stein-O'Brien et al., 2018*).

In this study, we performed comprehensive gene module *trans*-eQTL analysis across six major blood cell types and three stimulated conditions from five published datasets. To maximise gene module detection, we applied five distinct co-expression analysis methods (ICA, PEER, PLIER, WGCNA, funcExplorer) to the full dataset as well as individual cell types and conditions separately. Using a novel aggregation approach based on statistical fine mapping, we grouped individual *trans*-eQTLs to a set of non-overlapping loci. Extensive follow-up with gene set and transcription factor motif enrichment analyses allowed us to gain additional insight into the functional impact of *trans*-eQTLs and prioritise loci for further analyses. In addition to replicating two known monocyte-specific *trans*-eQTLs at the *IFNB1* (*Fairfax et al., 2014*; *Quach et al., 2016*; *Ramdhani et al., 2020*; *Ruffieux et al., 2018*) and *LYZ* loci (*Fairfax et al., 2012*; *Rakitsch and Stegle, 2016*; *Rotival et al., 2011*), we found that the *trans*-eQTL at the *ARHGEF3* locus detected in multiple whole blood datasets (*Mao et al., 2019*; *Nath et al., 2017*; *Rotival et al., 2011*; *Wheeler et al., 2019*) was highly specific to platelets in our analysis. Finally, we also detected a novel association at the *SLC39A8* locus that controlled a group of genes encoding zinc-binding proteins in LPS-stimulated monocytes.

## Results

### Cell types, conditions and samples

We used gene expression and genotype data from five previously published studies from three independent cohorts (*Fairfax et al., 2014*; *Fairfax et al., 2012*; *Kasela et al., 2017*; *Momozawa et al., 2018*; *Naranbhai et al., 2015*). The data consisted of CD4+ and CD8+ T cells (*Kasela et al., 2017*; *Momozawa et al., 2018*), B cells (*Fairfax et al., 2012*; *Momozawa et al., 2018*), neutrophils (*Momozawa et al., 2018*; *Naranbhai et al., 2015*), platelets (*Momozawa et al., 2018*), naive monoctyes (*Fairfax et al., 2014*; *Momozawa et al., 2018*) and monocytes stimulated with lipopolysaccharide for 2 or 24 hr (LPS 2 hr, LPS 24 hr) and interferon-gamma for 24 hr (IFNγ 24 hr) (*Fairfax et al., 2014*). The sample size varied from n = 226 in platelets to n = 710 in naive monocytes (*Figure 1A*).

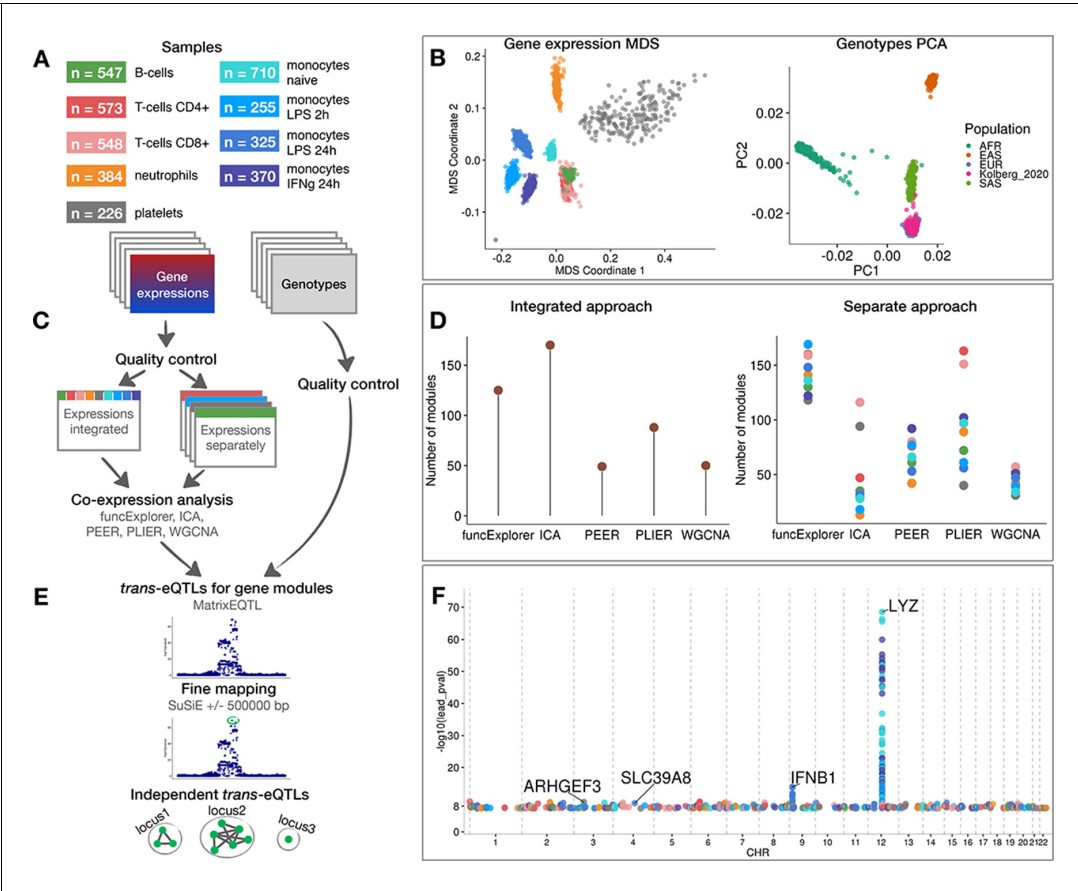

**Figure 1.** Data, analysis workflow and results. (A) Sample sizes of cell types and conditions included in the analysis. LPS - lipopolysaccharide, IFNg - interferon-gamma. (B) Multidimensional scaling (MDS) analysis of the gene expression data and principal component analysis (PCA) of genotype data after quality control and normalisation. Cell types and conditions are colour-coded according to panel A. Genotyped samples from this study have been projected to the 1000 Genomes Project reference populations. (C) Following quality control, five co-expression methods were applied to two different data partitioning approaches: (1) gene expression profiles across all cell types and conditions were analysed together (integrated approach), (2) gene expression profiles from each cell type and condition were analysed separately (separate approach). (D) The number of gene modules detected from integrated and separate analyses. (E) For *trans*-eQTL analysis we used the estimated module activity profile ('eigengene') as our phenotype. To identify independent *trans*-eQTLs, we performed statistical fine mapping for all nominally significant (p-value<5×10$^{-8}$) associations and grouped together all associations with overlapping credible sets. (F) Manhattan plot of nominally significant (p-value<5×10$^{-8}$) *trans*-eQTLs. Each point corresponds to a gene module that was associated with the corresponding locus and is colour-coded by the cell type from panel A.

The online version of this article includes the following figure supplement(s) for figure 1:

**Figure supplement 1.** Number of detected gene modules and their size distributions across co-expression analysis methods and data partitioning approaches.

**Figure supplement 2.** Aggregating fine mapping credible sets in the example of *ARHGEF3 trans*-eQTL locus.

**Figure supplement 3.** Monocytes-specific *trans*-eQTL near *IFNB1* that was first detected by *Fairfax et al., 2014* and later replicated in *Quach et al., 2016* after 6 hr of LPS stimulation (*Table 1*).

**Figure supplement 4.** Monocyte-specific *trans*-eQTL near *LYZ* that has been previously detected in three independent studies (*Fairfax et al., 2012*; *Rakitsch and Stegle, 2016*; *Rotival et al., 2011*; *Table 1*).

**Figure supplement 5.** The complete analysis workflow from gene co-expression analysis and gene module *trans*-eQTL mapping to multi-step filtering approaches.

**Figure supplement 6.** Simulation of eigenvectors obtained from different partitions of the same data versus the full data.

After quality control, normalisation and batch correction (see 'Materials and methods'), the final dataset consisted of 18,383 unique protein coding genes profiled in 3938 samples from 1037 unique genotyped individuals of European ancestries (*Figure 1B*). Even though the samples originated from five different studies, they clustered predominantly by cell type of origin (*Figure 1B*).

## Detecting *trans*-eQTLs regulating modules of co-expressed genes

We performed co-expression analyses with ICA, WGCNA, PLIER, PEER and funcExplorer on the full gene expression dataset (integrated approach) as well as on each cell type and condition separately (separate approach) (*Figure 1C*). In total, we obtained 482 gene modules from the integrated approach and 3509 from the separate clustering of different cell types (*Figure 1D*; *Figure 1—figure supplement 1*). For every module, the methods inferred a single characteristic expression pattern ('eigengene') that represents the expression profiles of the module genes across the samples. Although implementation details varied between methods (see 'Materials and methods'), these eigengene profiles were essentially linear combinations of expression levels of genes belonging to the modules.

The number of detected modules and their sizes varied due to the properties and the default parameters of each method (*Figure 1D*). Although matrix factorisation approaches generally identified larger modules than clustering methods (*Figure 1—figure supplement 1*), this is confounded by the fact that assigning genes to modules in matrix factorisation is fuzzy and requires the specification of arbitrary thresholds. Nevertheless, even though the number of modules for PEER and PLIER were initialised with identical values, PLIER consistently detected more modules with each module containing slightly fewer genes (*Figure 1D*, *Figure 1—figure supplement 1*). Similarly, funcExplorer detected more modules than WGCNA (*Figure 1D*) probably because funcExplorer was able to detect modules containing fewer genes (minimum of 5 *versus* 20 genes) if these were supported by functional enrichment (*Figure 1—figure supplement 1*).

For *trans*-eQTL analysis, we included 6,861,056 common (minor allele frequency >5%) genetic variants passing strict quality control criteria. First, we used linear regression implemented in Matrix-EQTL (*Shabalin, 2012*) package to identify all genetic variants nominally associated (p-value<$5 \times 10^{-8}$) with the eigengenes of each of the 3991 co-expression modules detected across nine cell types and conditions. We performed *trans*-eQTL analysis in each cell type and condition separately. Next, we used SuSiE (*Wang et al., 2018*) to fine map all significant associations to 864 independent credible sets of candidate causal variants (*Figure 1E*). Since we applied five co-expression methods to both integrated and cell-type-specific (separated) datasets, we found a large number of overlapping genetic associations. We thus aggregated overlapping credible sets from 864 associations to 601 non-overlapping genomic loci (*Figure 1—figure supplement 2*; see 'Materials and methods'). We observed that some, especially smaller, co-expression modules were driven by strong *cis*-eQTL effects that were controlling multiple neighbouring genes in the same module. To exclude such effects, we performed gene-level eQTL analysis for 18,383 protein-coding genes and the 601 lead variants identified above. We excluded co-expression modules where the module lead variant was not individually associated with any of the module genes in *trans* (>5 Mb away) and the overlap between the module genes and individually mapping *trans* genes was not significant according to the one-sided Fisher's exact test (Bonferroni adjusted p-value<0.05) (see 'Materials and methods'). This step reduced the number of nominally significant *trans*-eQTL loci to 247 (*Figure 1F*; *Supplementary files 1–2*). Finally, to account for the number of co-expression modules tested, we used both Benjamini-Yekutieli false discovery rate (BY FDR) and Bonferroni correction (see 'Materials and methods'). The BY FDR 10% threshold reduced the number of significant associations to 38 and Bonferroni threshold retained only three significant loci, including loci near *IFNB1* (*Figure 1—figure supplement 3*) and *LYZ* (*Figure 1—figure supplement 4*) genes that have been previously reported in several other studies (*Fairfax et al., 2014*; *Fairfax et al., 2012*; *Quach et al., 2016*; *Rakitsch and Stegle, 2016*; *Rotival et al., 2011*; *Ruffieux et al., 2018*; *Table 1*). While the strong *trans*-eQTL signals at the *IFNB1* and *LYZ* loci were detected by all co-expression methods in both integrated and separate analyses, most associations were detected by only a subset of the analytical approaches (*Supplementary file 1*).

To characterise the general interpretability of the associated modules, we performed functional enrichment analysis for all modules associated with the 247 nominally significant loci (*Supplementary file 3*). We found that 97% of the associated modules were enriched with at least one biological function from Gene Ontology, Reactome or KEGG. In contrast, in the gene-level analysis, only 86% of the loci showed significant enrichment in at least one tested cell type. However, this discrepancy could be partly due to the fact that gene-level analysis results in fewer associated genes, thus reducing the power to detect significant enrichments. Moreover, funcExplorer and PLIER

**Table 1.** Literature-based replication of *trans*-eQTL loci near *IFNB1*, *LYZ* and *ARHGEF3* genes.

Linkage disequilibrium ($r^2$) was calculated using European samples from the 1000 Genomes Phase 3 reference panel. The last column indicates if any of the associated modules had a significant overlap with the genes reported by the independent study according to one-sided Fisher's exact test after Bonferroni correction. The overlaps with individual modules are shown in **Supplementary file 5**. GHS - Gutenberg Health Study, FHS - Framingham Heart Study, CTS - Cardiogenics Transcriptomic Study, * - largest observed $r^2$ in the credible set.

| *trans*-eQTL | | | Replication | | | | | | |
|---|---|---|---|---|---|---|---|---|---|
| Locus | Lead rs ID | Context | Study | Dataset | Context | rs ID | $r^2$ | Replication variant in credible set | Significant overlap with a module |
| *IFNB1* | rs13296842 | Monocytes LPS 24 hr | *Fairfax et al., 2014* | Fairfax_2014 | Monocytes LPS 24 hr | rs2275888 | 0.57 (0.86*) | FALSE | - |
| | | | *Quach et al., 2016* | Quach_2016 | Monocytes LPS 6 hr | rs12553564 | 0.57 (0.86*) | FALSE | TRUE |
| | | | *Ramdhani et al., 2020* | Fairfax_2014 | Monocytes LPS 24 hr | rs2275888 | 0.57 (0.86*) | FALSE | - |
| | | | *Ruffieux et al., 2018* | Fairfax_2014 | Monocytes LPS 24 hr | rs3898946 | 0.88 | TRUE | - |
| *LYZ* | rs10784774 | Monocytes naive, LPS 2 hr, LPS 24 hr, IFNγ 24 hr | *Rotival et al., 2011* | GHS | Monocytes | rs11177644 | 0.79 | TRUE | TRUE |
| | | | *Fairfax et al., 2012* | Fairfax_2012 | Monocytes | rs10784774 | 1 | TRUE | - |
| | | | *Rakitsch and Stegle, 2016* | CTS | Monocytes | rs6581889 | 0.79 | TRUE | TRUE |
| *ARHGEF3* | rs1354034 | Platelets | *Võsa et al., 2018* | eQTLGen | Blood | rs1354034 | 1 | TRUE | TRUE |
| | | | *Mao et al., 2019* | Battle_2014 | Blood | rs1354034 | 1 | TRUE | - |
| | | | *Rotival et al., 2011* | GHS | Monocytes | rs12485738 | 0.6 | FALSE | - |
| | | | | | | rs1344142 | 0.6 | TRUE | - |
| | | | *Wheeler et al., 2019* | FHS | Blood | - | - | - | - |
| | | | *Nath et al., 2017* | DILGOM07 | Blood | rs1354034 | 1 | TRUE | TRUE |

modules are based on known gene annotations and are therefore expected to have high levels of enrichment by definition. We will now dissect two loci with interesting functional enrichment patterns in more detail.

## Platelet-specific *trans*-eQTL at the *ARHGEF3* locus is associated with multiple platelet traits

We found that the rs1354034 (T/C) variant located within the *ARHGEF3* gene is associated with three co-expression modules in platelets: one ICA module detected in integrated analysis (IC68, 1074 genes) and two co-expression modules detected in a platelet-specific analysis by PLIER (X6. WIERENGA_STAT5A_TARGETS_DN, 918 genes) and funcExplorer (Cluster_12953, five genes) (**Figure 2B**, **Figure 2—figure supplement 1**). The T allele increases the expression of the *ARHGEF3* gene in *cis* and the two lead variants are the same (**Figure 2A**). Furthermore, both the *cis* and *trans*-eQTLs colocalise with a GWAS hit for mean platelet volume (*cis* PP4 = 0.99, *trans* PP4 >0.99 for all modules), platelet count (*cis* PP4 = 0.99, *trans* PP4 >0.99 for all modules) and plateletcrit (*trans* PP4 >0.99 for all modules) (**Figure 2A**; **Astle et al., 2016**). Interestingly, *ARHGEF3* itself is not in any of the three modules and the module eigengenes are not strongly co-expressed with *ARHGEF3* (Pearson's r ranging from 0.07 to 0.33 in platelets). While IC68 and X6.WIERENGA_STAT5A_TAR-GETS_DN share 74 overlapping genes (one-sided Fisher's exact test p-value=0.003), none of the genes in Cluster_12953 is in any of the other modules.

Although the *ARHGEF3 trans*-eQTL has been detected in multiple whole blood *trans*-eQTL studies (**Mao et al., 2019**; **Nath et al., 2017**; **Võsa et al., 2018**; **Wheeler et al., 2019**; **Table 1**), our analysis demonstrates that this association is highly specific to platelets and not detected in other

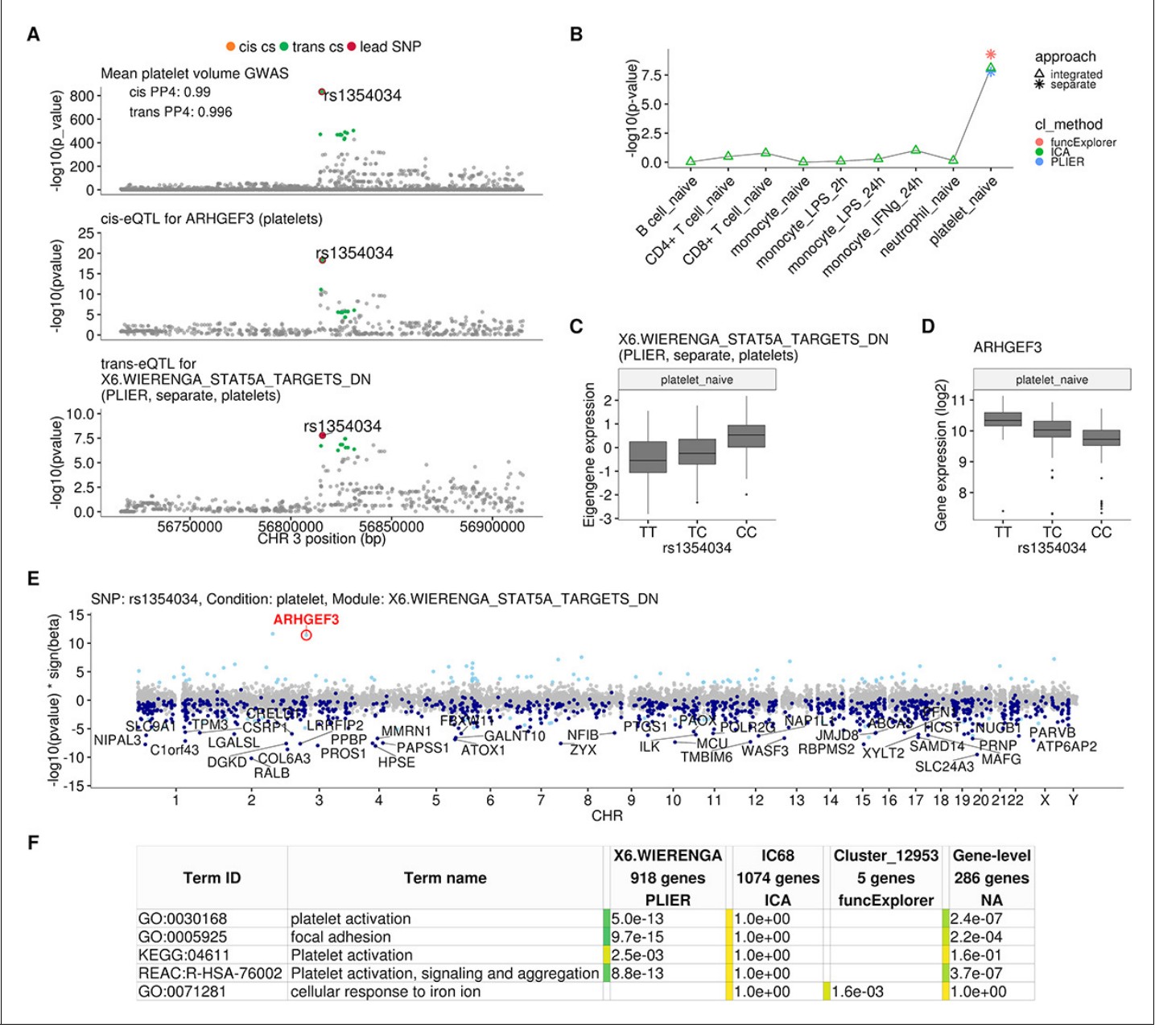

**Figure 2.** Platelet-specific *trans*-eQTL at the *ARHGEF3 locus.* (A) Regional plots showing colocalisation between GWAS signal for mean platelet volume (*Astle et al., 2016*), *cis*-eQTL for *ARHGEF3* in platelets and *trans*-eQTL for a platelet-specific co-expression module detected by PLIER. *Cis* and *trans* credible sets (cs) are marked on the plots. The *cis* credible set consists of only the lead variant (rs1354034), which occludes the orange highlight. (B) Line graph showing that the association between the modules and *ARHGEF3* locus is platelet specific. In cell-type-specific clustering, only a single p-value from the corresponding cell type is available. The integrated modules have p-values from each of the cell types and the values are connected by a line. (C) Association between the *trans*-eQTL lead variant (rs1354034) and eigengene of module X6.WIERENGA_STAT5A_TARGETS_DN in platelets. (D) Association between the *trans*-eQTL lead variant (rs1354034) and *ARHGEF3* expression in platelets. (E) Manhattan plot of gene-level eQTL analysis for the *trans*-eQTL lead variant. Dark blue points highlight the genes in module X6.WIERENGA_STAT5A_TARGETS_DN. Light blue points show significantly associated genes (variant-level Benjamini-Hochberg FDR 5%) not included in the module. (F) Functional enrichment analysis of modules associated with *ARHGEF3* locus (see full results at https://biit.cs.ut.ee/gplink/l/CY6ZukXhSq). Empty cell indicates that no gene in the module is annotated to the corresponding term, enrichment p-value=1 shows that at least some of the genes in the module are annotated to the term, but not enough to report over-representation. The last column combines the FDR 5% significant genes from the gene-level analysis. The table shows adjusted enrichment p-values. GO - Gene Ontology, KEGG - Kyoto Encyclopedia of Genes and Genomes Pathways, REAC - Reactome Pathways.

The online version of this article includes the following figure supplement(s) for figure 2:

**Figure supplement 1.** Intronic variant located within the *ARHGEF3* gene (rs1354034) is associated with three co-expression modules in platelets.

*Figure 2 continued*

**Figure supplement 2.** Replication of the *ARHGEF3 trans*-eQTL association in the eQTLGen study (*Võsa et al., 2018*).

**Figure supplement 3.** Mediation analysis between the *cis*-eQTL for *ARHGEF3* and *trans*-eQTLs for three co-expression modules associated with the same lead variant (rs1354034).

major blood cell types (*Figure 2B*). Furthermore, even though *ARHGEF3* is expressed in multiple cell types, the *cis*-eQTL effect is also only visible in platelets (*Figure 2—figure supplement 1*). Reassuringly, the *trans*-eQTL effect sizes in our small platelet sample (n = 216) are correlated (Pearson's r = 0.68, p-value=$5.1 \times 10^{-12}$) with the effects from the largest whole blood *trans*-eQTL meta-analysis (*Võsa et al., 2018*) (n = 31,684) (*Figure 2—figure supplement 2*). The platelet specificity of the *ARHGEF3* association is further supported by functional enrichment analysis with g:Profiler (*Raudvere et al., 2019*), which found that both the PLIER module X6.WIERENGA_STAT5A_TARGETS_DN and target genes from the gene-level analysis were strongly enriched for multiple terms related to platelet activation (*Figure 2E*; https://biit.cs.ut.ee/gplink/l/CY6ZukXhSq). Cluster_12953, however, was enriched for cellular response to iron ion, suggesting that *ARHGEF3* might be involved in multiple independent processes (*Mao et al., 2019*; *Serbanovic-Canic et al., 2011*). Altogether, these results demonstrate how a *trans*-eQTL detected in whole blood can be driven by a strong signal present in only one cell type.

## *SLC39A8* locus is associated with zinc ion homeostasis in LPS-stimulated monocytes

One of the novel results in our analysis was a locus near the *SLC39A8* gene that was associated (p-value=$1.2 \times 10^{-9}$) with a single co-expression module detected by funcExplorer (Cluster_10413) in monocytes stimulated with LPS for 24 hr (*Figure 3A–C*). The module consisted of five metallothionein genes (*MT1A*, *MT1F*, *MT1G*, *MT1H*, *MT1M*) all located in the same locus on chromosome 16 (*Figure 3D*). Although the *trans*-eQTL lead variant (rs75562818) was significantly associated with the expression of the *SLC39A8* gene (*Figure 3A and D*), the two association signals did not colocalise and the credible sets did not overlap (*Figure 3A*; *Figure 3—figure supplement 1*), indicating that the *cis*-eQTL detected in naive and stimulated monocytes in our dataset is not the main effect driving the *trans*-eQTL signal. Furthermore, the expression of *SLC39A8* was only moderately correlated with the eigengene value of Cluster_10413 (Pearson's r = 0.27). Since *SLC39A8* is strongly upregulated ($\log_2$fold-change = 3.53) in response to LPS already at 2 hr (*Figure 4A*), we speculated that there might be a transient eQTL earlier in the LPS response. To test this, we downloaded the *cis*-eQTL summary statistics from the *Kim-Hellmuth et al., 2017* study that had mapped eQTLs in monocytes stimulated with LPS for 90 min and 6 hr (*Kim-Hellmuth et al., 2017*). Indeed, we found that the *cis*-eQTL 90 min after LPS stimulation colocalised with our *trans*-eQTL (*Figure 3A*) and this signal disappeared by 6 hr after stimulation (*Figure 3—figure supplement 2*).

To understand the function of the *SLC39A8* locus, we turned to the target genes. Gene-level analysis identified two more metallothionein genes (*MT1E* and *MT1X*) from the same locus as likely target genes (*Figure 3D*). Enrichment analysis with g:Profiler revealed that these genes were enriched for multiple Gene Ontology terms and pathways related to zinc ion homeostasis (*Figure 3E*, full results at https://biit.cs.ut.ee/gplink/l/aohV4uKeT1). Furthermore, the promoter regions of the seven genes were also enriched for the binding motif of the metal transcription factor 1 (MTF1) transcription factor (p-value=$2.1 \times 10^{-4}$, *Figure 3E*). Taken together, these results suggest that a transient eQTL of the *SLC39A8* gene 90 min after stimulation regulates the expression of 7 zinc-binding proteins 24 hr later. Multiple lines of literature evidence support this model (*Figure 4B*). First, the ZIP8 protein coded by the *SLC39A8* gene is a manganese and zinc ion influx transporter (*Nebert and Liu, 2019*). Secondly, *SLC39A8* is upregulated by the NF-κB transcription factor in macrophages and monocytes in response to LPS and this upregulation leads to increased intracellular $Zn^{2+}$ concentration (*Liu et al., 2013*). Third, $Zn^{2+}$ influx increases the transcriptional activity of the metal transcription factor 1 (MTF1) (*Kim et al., 2014*) and metallothioneins, which act as $Zn^{2+}$-storage proteins, are well known target genes of the MTF1 transcription factor (*Laity and Andrews, 2007*). Finally, *SLC39A8* knockdown in mice leads to decreased expression of the metallothionein 1 (*MT1*) gene (*Liu et al., 2013*).

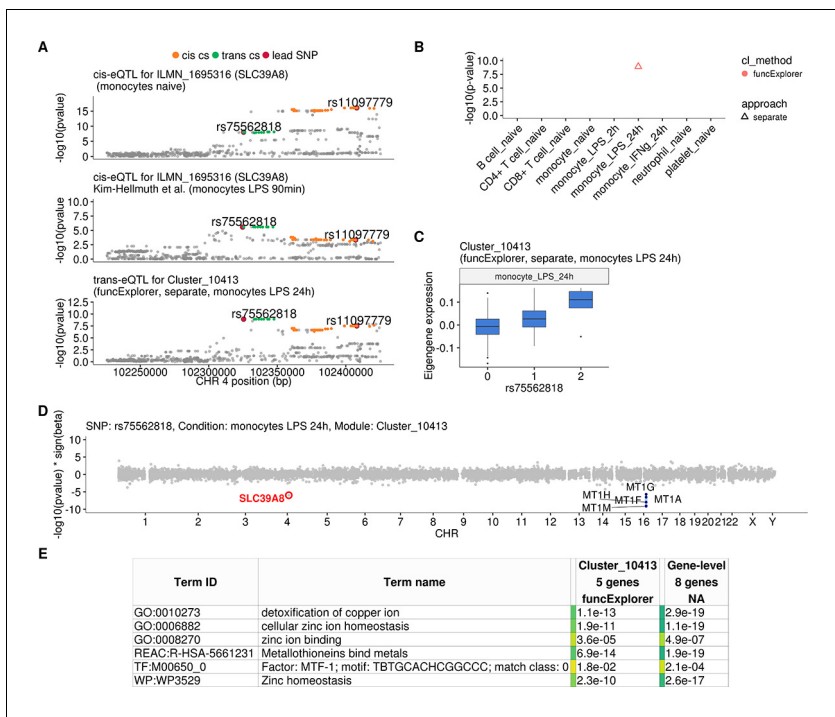

**Figure 3.** Transient *cis*-eQTLs for *SLC39A8* is associated with the expression of seven metallothionein genes in *trans* in monocytes stimulated with LPS for 24 hr. (**A**) Regional plots comparing association signals between naive (rs11097779) and transiently induced *cis*-eQTLs (rs75562818) for *SLC39A8* and *trans*-eQTL (rs75562818) for a module of five co-expressed metallothionein genes. LPS-induced *cis*-eQTL summary statistics 90 min post stimulation (n = 134) were obtained from *Kim-Hellmuth et al., 2017*. (**B**) Graph showing that the association between the module and *SLC39A8* locus is stimulation specific. As this module was detected by a cell-type-specific clustering, only a single value from the corresponding cell type is available. (**C**) Association between *trans*-eQTL (rs75562818) and eigengene of funcExplorer module Cluster_10413 in monocytes after 24 hr of LPS stimulation. (**D**) Manhattan plot of gene-level eQTL analysis for rs75562818. Dark blue points highlight the genes in module Cluster_10413. Light blue points show significantly associated genes (variant-level Benjamini-Hochberg FDR 5%) not included in the module. (**E**) Functional enrichment analysis of the *SLC39A8* associated module (see https://biit.cs.ut.ee/gplink/l/aohV4uKeT1 for full results). The last column combines the FDR 5% significant genes from the gene-level analysis. The table shows adjusted enrichment p-values. MTF1 - metal transcription factor 1. GO - Gene Ontology, WP - WikiPathways, REAC - Reactome Pathways, TF - transcription factor binding sites from TRANSFAC.

The online version of this article includes the following figure supplement(s) for figure 3:

**Figure supplement 1.** The *SLC39A8 trans*-eQTL lead variant (rs75562818) and *cis*-eQTLs for *SLC39A8* probe sets (ILMN_1695316 and ILMN_2233539) do not colocalise in any of the monocyte conditions in our data.

**Figure supplement 2.** Regional plots comparing association signals of *SLC39A8 trans*-eQTL (rs75562818) and *cis*-eQTLs from *Kim-Hellmuth et al., 2017*.

**Figure supplement 3.** Mediation analysis between the *cis*-eQTL for *SLC38A8* and the *trans* module associated with the same lead variant (rs75562818) in monocytes stimulated with LPS for 24 hr.

To see if the *SLC39A8 trans*-eQTL might be associated with any higher level phenotypes, we queried the GWAS Catalog database (*Buniello et al., 2019*) with the ten variants from the *trans*-eQTL 95% credible set. We found that a lead variant for red blood cell distribution width (rs7692921) was one of the variants in our credible set and in high LD ($r^2$ = 0.991) with the *trans*-eQTL lead variant (*Figure 4C*; *Kichaev et al., 2019*). However, neither of the eQTL variants was in LD with a known missense variant (rs13107325) in the *SLC39A8* gene that has been associated with schizophrenia, Parkinson's disease and other traits (*Figure 4C*; *Pickrell et al., 2016*).

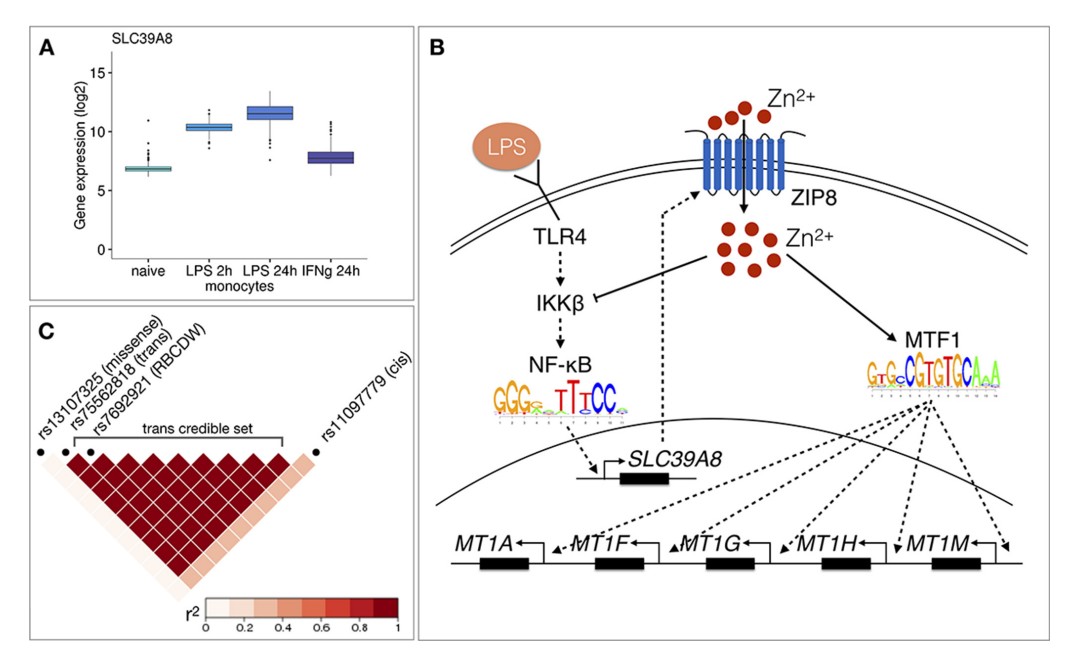

**Figure 4.** Molecular mechanisms underlying the *SLC39A8 trans*-eQTL locus. (A) *SLC39A8* gene expression values (log$_2$ intensities) across naive and stimulated monocytes. (B) Overview of the known regulatory interactions underlying the *cis* and *trans* eQTL effects at the *SLC39A8* locus. Figure adapted from *Liu et al., 2013*. (C) Pairwise LD (r$^2$ within 1000 Genomes European populations) between the *SLC39A8* variants highlighting missense variant (rs13107325), *trans*-eQTL (rs75562818), red blood cell distribution width (RBCDW) associated SNP (rs7692921) in our credible set and the *cis* lead variant from naive monocytes (rs11097779). LD was calculated using the LDlinkR (v.1.0.2) R package (*Myers et al., 2020*).

## Mediation analysis

For three of the four *trans*-eQTL loci discussed above (*LYZ*, *ARHGEF3* and *SLC39A8*), we also detected an overlapping *cis*-eQTL effect on one or more *cis* genes. To test if the *cis*-eQTL effect might mediate the observed *trans* effect on the co-expression modules, we used mediation analysis. In all three cases, we detected a statistically significant mediation effect between the *cis* and *trans* associations (*Figure 2—figure supplement 3*, *Figure 3—figure supplement 3*, *Supplementary file 4*). However, in all cases, the mediation explained only a small fraction of the total genotype effect on the co-expression module. There could be multiple reasons for this. First, since co-expression module eigengene values go through multiple transformations, this might introduce additional noise and thus reduce observed mediation effect (*Pierce et al., 2014*). Second, if there is a temporal delay between the *cis* and *trans* effects (as observed for *SLC39A8*) then we would not necessarily expect to detect mediation at the same time point, even if the *cis* eQTL is causal for the *trans* eQTL effect. Finally, multiple independent causal variants in the region that are in LD with each other could bias the mediation estimates (*Figure 3A*).

## Replication of associations in independent datasets

We first performed a literature-based replication to measure the overlap between the modules that map to the loci near *IFNB1*, *LYZ* and *ARHGEF3* with the genes reported by previous studies (*Table 1*, *Supplementary file 5*). All the modules associated with the *IFNB1* locus in monocytes stimulated with LPS for 24 hr (12 in total) had a significant overlap (one-sided Fisher's exact test, Bonferroni adjusted p-value<0.05) with the *trans* genes reported by *Quach et al., 2016*. At the *LYZ* locus, we compared the 30 modules detected in unstimulated monocytes with the *trans* genes reported by *Rotival et al., 2011* and *Rakitsch and Stegle, 2016*. In the case of Rotival et al., 23 out of the 30 modules from our study had significant overlap with the 33 *trans* genes reported by Rotival et al. In contrast, only two of our modules had a significant overlap with the genes reported by Rakitsch and Stegle. Interestingly, only one *trans* associated gene was shared between Rotival et al. and (*Rakitsch and Stegle, 2016*). We also evaluated the overlap for the three modules associated with

the *ARHGEF3* locus and the 840 genes reported in the eQTLGen study (*Võsa et al., 2018*). Only one module, IC68, did not have a significant overlap but this could be due to its large size and the fuzzy definition of the ICA module membership. For *ARHGEF3* we also compared the modules with the 163 *trans* genes reported by *Nath et al., 2017* where only one module (X6.WIERENGA_STAT5A_TARGETS_DN) had a significant overlap.

To further assess the replication of identified *trans*-eQTLs (after filtering for Benjamini-Yekutieli FDR<10%), we compared associated modules in unstimulated monocytes, neutrophils and T-cells to matched cell types from three independent studies for which we had access to individual-level data: BLUEPRINT (*Chen et al., 2016*), ImmVar (*Raj et al., 2014*) and *Quach et al., 2016*. We analysed 9 of the 38 *trans*-eQTLs that were associated with 40 different modules. We compared the overlap of gene modules and corresponding significant gene-level results (variant-level FDR <5%) from these three independent studies. Unfortunately, we were not able to replicate any additional associations. Interestingly, even though the *LYZ and YEATS4 cis*-eQTL effect was present in all three studies, the *trans*-eQTL did not replicate in any of them. Since this *trans*-eQTL was previously detected by Rotival et al, this suggests that in addition to small sample sizes of the replication studies, there might be biological differences in how the samples were collected.

## Discussion

Given that *trans*-eQTLs have been more difficult to replicate between studies and false positive associations can easily occur due to technical issues (*Dahl et al., 2019*; *Saha and Battle, 2018*), it is increasingly important to effectively summarise and prioritise associations for follow-up analyses and experiments. We found that aggregation of credible sets of eigengene profiles from multiple co-expression methods (*Figure 1—figure supplement 2*) successfully reduced the number of independent associations, but this still retained 243 loci that we needed to evaluate. To further prioritise associations, we used gene set and transcription factor motif enrichment analysis of the *trans*-eQTL target genes. Although motif analysis is often underpowered, it can provide directly testable hypotheses about the *trans*-eQTL mechanism such as the MTF1 transcription factor that we identified at the *SLC39A8* locus. Similar approaches have also been successfully used to characterise *trans*-eQTLs involving IRF1 and IRF2 transcription factors (*Brandt et al., 2020*; *Fairfax et al., 2014*).

A major limitation of co-expression-based approach for *trans*-eQTL mapping is that many true co-expression modules can remain undetected by various co-expression analysis methods (*Way et al., 2020*). We sought to overcome this by aggregating results across five complementary co-expression methods. We found that while all methods were able to discover strong co-expression module *trans*-eQTLs such as those underlying the *IFNB1* (*Figure 1—figure supplement 3*) and *LYZ* (*Figure 1—figure supplement 4*) associations, most co-expression module *trans*-eQTLs were only detected by a subset of the analysis methods. For example, the *ARHGEF3* association was detected by three of the five methods (*Figure 2B*) and *SLC39A8* co-expression module was found only by funcExplorer and only when samples from LPS-stimulated monocytes were analysed separately (*Figure 3B*). Since this module consisted of only seven strongly co-expressed genes, other methods were probably not well tuned to find it. Moreover, if the *trans*-eQTL locus controls a single or a small number of genes then co-expression-based approaches are probably not well suited to detect such associations and gene-level analysis is still required.

To maximise module discovery, we aggregated results from five co-expression analysis methods and two partitions of the same underlying data (integrated *versus* separate). While this reduced the number of tests compared to a standard gene-level analysis, it introduced an additional layer of complexity, because the same gene expression values contributed to multiple different co-expression modules and analytical settings. As a result, it is unclear how well calibrated our false discovery rate estimates are. Thus, we decided to first use a relaxed nominal significance threshold of p-value<$5 \times 10^{-8}$, assuming that most of those associations were likely to be false positives. In our subsequent follow-up analyses, we only focused on four loci that we could either replicate in independent datasets (*IFNB1*, *LYZ*, *ARHGEF3*) or find significant support from the literature (*SLC39A8*).

Since eQTL datasets from purified cell types are still relatively small and single-cell eQTL datasets are even smaller (*van der Wijst et al., 2018*), it is tempting to perform *trans*-eQTL analysis on whole tissue datasets such as the brain or whole blood (*Võsa et al., 2018*). However, it remains unclear what fraction of cell type and condition-specific *trans*-eQTLs can be detected in whole tissue

datasets collected from healthy donors. Although we were able to replicate the *ARHGEF3* association in the eQTLGen whole blood meta-analysis, because our fine mapped lead variant happened to be one of the 10,317 variants tested in eQTLGen, systematic replication requires genome-wide summary statistics that are currently lacking for *trans*-eQTL analyses. Secondly, tissue datasets can be biased by cell type composition effects. These can lead to spurious *trans*-eQTL signals, because genetic variants associated with cell type composition changes would appear as *trans*-eQTLs for cell-type-specific genes (*Võsa et al., 2018*). Furthermore, multiple studies have demonstrated that the co-expression signals in tissues are also largely driven by cell type composition effects (*Farahbod and Pavlidis, 2019*; *Parsana et al., 2019*; *Schubert et al., 2020*). Thus, even though PLIER detected the *ARHGEF3* *trans*-eQTL in whole blood, this could have been at least partially driven by the change in platelet proportion between individuals (*Mao et al., 2019*). Our analysis in purified cell types enabled us to verify that this was a truly platelet-specific genetic association.

Although both in the case of *ARHGEF3* and *SLC39A8,* we detected significant mediation between the expression level of the *cis* gene and the observed *trans*-eQTL effect, it explained only a small proportion of the total *trans* effect. Furthermore, there was only a modest correlation (Pearson's r between 0.07 and 0.33) between the *cis* gene expression and the corresponding *trans* co-expression module expression. In case of *SLC39A8* there seemed to be a temporal delay with the *cis*-eQTL being active early in LPS response and *trans*-eQTL appearing much later after proposed accumulation of the ZIP8 protein and increase in intracellular zinc concentration. Temporal delay has similarly been reported for the *trans*-eQTLs at the *INFB1* (*Fairfax et al., 2014*) and *IRF1* (*Brandt et al., 2020*) loci. This suggests that if *cis* and *trans* effects are separated from each other either in time (early *versus* late response) or space (different cell types that interact with each other), then this might limit the power of methods that rely on genetically predicted gene expression levels to identify regulatory interactions (*Liu et al., 2018*; *Luijk et al., 2018*; *Wheeler et al., 2019*) and infer causal models. This can also have a negative impact on mediation analysis (*Battle et al., 2014*; *Chick et al., 2016*; *Yang et al., 2019*), which seeks to estimate the proportion of *trans*-eQTL variance explained by the expression level of the *cis* gene. Altogether, our results indicate that limiting *trans*-eQTL analysis to missense variants and to variants that have been detected as *cis*-eQTLs in the same cell type might miss some true associations, because the *cis* effect might be active in some other, yet unprofiled, context.

We have performed a large-scale *trans*-eQTL analysis in six blood cell types and three stimulated conditions. We demonstrate that co-expression module detection combined with gene set enrichment analysis can help to identify interpretable *trans*-eQTLs, but these results depend on which co-expression method is chosen for analysis and how the input data are partitioned beforehand. We perform in-depth characterisation of two cell type specific *trans*-eQTL loci: platelet-specific *trans*-eQTL near the *ARHGEF3* gene and monocyte-specific associations near the *SLC39A8* locus. In both cases, the co-expression modules were enriched for clearly interpretable Gene Ontology terms and pathways, which directly guided literature review and more detailed analyses. We believe that applying co-expression and gene set enrichment based approaches to larger eQTL datasets has the power to detect many more additional associations while simultaneously helping to prioritise *trans*-eQTLs for detailed experimental or computational characterisation. A particularly promising avenue would be treating co-expression modules as complex traits for which multiple independent genetic associations could be mapped. These associations could subsequently be used in Mendelian randomisation analyses to infer causal intermediate phenotype for complex diseases (*Evans and Davey Smith, 2015*).

## Materials and methods

### Datasets used in the analysis
CEDAR

The CEDAR dataset (*Momozawa et al., 2018*) contained gene expression and genotype data from CD4+ T-cells, CD8+ T-cells, CD19+ B-cells and CD14+ monocytes, CD15+ neutrophils and platelets from up to 323 individuals. The raw gene expression data generated with Illumina HumanHT-12 v4 arrays were downloaded from ArrayExpress (*Athar et al., 2019*) (accession E-MTAB-6667). The raw

IDAT files were imported into R using the readIdatFiles function from the beadarray v2.28 (*Dunning et al., 2007*) Bioconductor package.

The raw genotype data generated by Illumina HumanOmniExpress-12 v1_A genotyping arrays were also downloaded from ArrayExpress (accession E-MTAB-6666). Genotype calling was performed with Illumina GenomeStudio v2.0.4, after which the raw genotypes were exported in PLINK format.

### Kasela et al., 2017

*Kasela et al., 2017* generated gene expression and genotype data from CD4+ and CD8+ T cells from 297 unique donors. The raw gene expression data generated with Illumina HumanHT-12 v4 arrays were downloaded from Gene Expression Omnibus (accession GSE78840). The genotype data generated by Illumina HumanOmniExpress-12 v1_A genotyping arrays were obtained from the Estonian Genome Center, University of Tartu (https://genomics.ut.ee/en/access-biobank). Ethical approval was obtained from the Research Ethics Committee of the University of Tartu (approval 287/T-14).

### Fairfax et al., 2012, Fairfax et al., 2014 and Naranbhai et al., 2015

*Fairfax et al., 2012* profiled gene expression in CD19+ B cells from 282 individuals (ArrayExpress accession E-MTAB-945). *Fairfax et al., 2014* profiled gene expression in naive CD14+ monocytes as well as in cells stimulated with lipopolysaccharide (LPS) for 2 or 24 hr and interferon-gamma for 24 hr from up to 414 individuals (accession E-MTAB-2232). *Naranbhai et al., 2015* profiled gene expression in CD15+ neutrophils from 93 individuals (accession E-MTAB-3536). The genotype data for all three studies were generated by Illumina HumanOmniExpress-12 genotyping arrays and were downloaded from European Genome-phenome Archive (accessions EGAD00010000144 and EGAD00010000520).

## Genotype data quality control and imputation

We started with raw genotype data from each study in PLINK format and GRCh37 coordinates. Before imputation, we performed quality control independently on each of the three datasets. Briefly, we used Genotype harmonizer (*Deelen et al., 2014*) v1.4.20 to align the alleles with the 1000 Genomes Phase 3 reference panel and excluded variants that could not be aligned. We used PLINK v1.9.0 to convert the genotypes to VCF format and used the fixref plugin of the bcftools v1.9 to correct any strand swaps. We used 'bcftools norm −`check-ref` x' to remove any remaining variants where the reference allele did not match the GRCh37 reference genome. Finally, we excluded variants with Hardy-Weinberg equilibrium p-value>$10^{-6}$, missingness >0.05 and MAF <0.01. We also excluded samples with more than 95% of the variants missing. Finally, we merged genotype data from all three studies into a single VCF file.

After quality control, we included 580,802 autosomal genetic variants from 1041 individuals for imputation. We used a local installation of the Michigan Imputation Server v1.2.1 (*Das et al., 2016*) to perform phasing and imputation with EAGLE v2.4 (*Loh et al., 2016*) and Minimac4 (*Das et al., 2016*). After imputation, we used CrossMap.py v2.8.0 (*Zhao et al., 2014*) to convert genotype coordinates to GRCh38 reference genome. We used bcftools v1.9.0 to exclude genetic variants with imputation quality score $R^2$ <0.4 and minor allele frequency (MAF) <0.05. We used PLINK (*Chang et al., 2015*) v1.9.0 to perform LD pruning of the genetic variants and LDAK (*Speed et al., 2012*) to project new samples to the principal components of the 1000 Genomes Phase 3 reference panel (*Sudmant et al., 2015*). The Nextflow pipelines for genotype processing and quality control are available from GitHub (https://github.com/eQTL-Catalogue/genotype_qc).

## Detecting sample swaps between genotype and gene expression data

We used Genotype harmonizer (*Deelen et al., 2014*) v1.4.20 to convert the imputed genotypes into TRITYPER format. We used MixupMapper (*Westra et al., 2011*) v1.4.7 to detect sample swaps between gene expression and genotype data. We detected 155 sample swaps in the CEDAR dataset, most of which affected the neutrophil samples. We also detected one sample swap in the *Naranbhai et al., 2015* dataset.

## Gene expression data quality control and normalisation

As a first step, we performed multidimensional scaling (MDS) and principal component analysis (PCA) on each dataset separately to detect and exclude any outlier samples. This was done after excluding the replicate samples and the samples that did not pass the genotype data quality control. Additional outliers were detected after quantile normalisation and adjusting for batch effects. The normalisation was performed using the lumiN function from the lumi v.2.30.0 R package (*Du et al., 2008*). Batch effects, where applicable, were adjusted for with the removeBatchEffect function from the limma v.3.34.9 R package (*Ritchie et al., 2015*). After quality control to exclude outlier samples, the quantile normalised $\log_2$ intensity values from all datasets were combined. This was followed by regressing out dataset specific batch effects. Only the intensities of 30,353 protein-coding probes were used. Finally, the probe sets were mapped to genes. For genes with more than one corresponding probe set, the probe with the highest average expression was used. 18,383 protein-coding genes with unique Ensembl identifiers remained for co-expression analysis. We did not regress out any principal components from the gene expression data, as this can introduce false positives in *trans*-eQTL analysis due to collider bias (*Dahl et al., 2019*). In total, 3938 samples remained after the quality control (*Table 2*).

## Co-expression analysis

We applied five different methods to identify modules of co-expressed genes from the gene expression data. We used an expression matrix where rows correspond to genes and columns to individuals/samples as input for the methods. The gene expression profiles were centred and standardised prior to analysis. All methods infer gene co-expression modules, each of which can be described by a single expression profile ('eigengene') that captures the collective behaviour of corresponding genes in the module. Although the eigengenes are defined differently for matrix factorisation and co-expression clustering methods, there is a straightforward connection between the two definitions (see below more details). These eigengenes are treated as quantitative traits in the *trans*-eQTL analysis. To detect potential cell type and condition-specific modules, we applied the same methods also to the expression matrices from each of the nine cell types and conditions separately. To reduce complexity, we relied on default parameters recommended by the authors of each co-expression analysis method. Exceptions to this are stated below. Summaries of the co-expression analysis results from both integrated and cell-type-specific expression data are shown in *Figure 1—figure supplement 1*.

## Co-expression clustering methods

### Weighted gene co-expression network analysis (WGCNA)

The WGCNA method (*Langfelder and Horvath, 2008*) identifies non-overlapping co-expressed gene modules. Each of the modules is represented by its first principal component of expression values of genes in the module termed as module eigengene. We used the function blockwiseModules for automatic block-wise network construction and module identification with default parameters

**Table 2.** Number of samples included in the analysis from each study and each cell type.

| Cell type | Fairfax_2012 | Fairfax_2014 | Naranbhai_2015 | Kasela_2017 | CEDAR |
|---|---|---|---|---|---|
| B cell | 281 | - | - | - | 266 |
| T cell CD4+ | - | - | - | 279 | 294 |
| T cell CD8+ | - | - | - | 267 | 281 |
| Neutrophil | - | - | 93 | - | 291 |
| Platelet | - | - | - | - | 226 |
| Monocyte naive | - | 420 | - | - | 290 |
| Monocyte LPS 2 hr | - | 255 | - | - | - |
| Monocyte LPS 24 hr | - | 325 | - | - | - |
| Monocyte IFNγ 24 hr | - | 370 | - | - | - |

from the dedicated R package WGCNA (v.1.66). The number of modules was detected automatically by the algorithm, but the number of genes in a module was limited between 20 and 5000 genes.

## funcExplorer

FuncExplorer (*Kolberg et al., 2018*) is a web tool that performs hierarchical clustering on gene expression values which is followed by automated functional enrichment analysis to derive the most biologically meaningful gene modules from the dendrogram. The expression data were uploaded to funcExplorer and the modules were detected using the following parameters: best annotation strategy, p-value threshold 0.01 for enrichment of Gene Ontology, KEGG and Reactome annotations. All other parameters were left to default values. Every funcExplorer gene module is characterised by the eigengene profile which, like in WGCNA, is the first principal component of module expression values calculated in the same way as in WGCNA. The number of modules is detected automatically by funcExplorer and the different modules consist of non-overlapping sets of genes, the default parameters limited module sizes between 5 and 1000 genes. The co-expression analysis results are available for browsing from https://biit.cs.ut.ee/funcexplorer/user/2a29dfa6de6b8b733f665352735a-daf5 where the option 'Dataset' includes the full selection of expression data used in this analysis. Dataset 'Merged_ENSG_expression' incorporates integrated samples from all cell types and conditions, 'CL_0000233_naive' stands for platelets, 'CL_0000236_naive' for B cells, 'CL_0000624_naive' for CD4+ T cells, 'CL_0000625_naive' for CD8+ T cells, 'CL_0000775_naive' for neutrophils, 'CL_0002057_naive' for monocytes and 'CL_0002057_IFNg_24_hr', 'CL_0002057_LPS_24 hr', 'CL_0002057_LPS_2 hr' include gene expression matrices from corresponding stimulated monocyte samples.

## Matrix factorisation methods

Matrix factorisation methods, such as ICA, PLIER and PEER, deconvolve the input gene expression matrix into two related matrices (*Stein-O'Brien et al., 2018*). One of the matrices is the matrix of factor loadings for each sample and the other describes the gene-level weights of the factors. In the case of ICA, PLIER and PEER, we used the factor loadings as module eigengene profiles. For g:Profiler enrichment analysis we used the gene-level weights to define the genes that characterise the modules by choosing the ones that are the most influenced, that is the genes at both extremes of gene weight values (two standard deviations from the mean weights in this module). Thus, different modules can include overlapping sets of genes.

### Independent component analysis (ICA)

The ICA (*Hyvärinen and Oja, 2000*) method attempts to decompose gene expression measurements into independent components (factors) which represent underlying biological processes. The fastICA (*Marchini et al., 2013*) algorithm in R was run using the runICA function from the wrapper package picaplot v.0.99.7 (https://github.com/jinhyunju/picaplot). The number of components to be estimated was automatically detected by the implementation using a 70% variance cut-off value and maximum of 10 iterations (set with parameters var_cutoff = 70 and max_iter = 10). The ICA algorithm was run 15 times (n_runs = 15) with five cores (n_cores = 5) and only the components that replicated in every run were returned by the function. All other parameters were left to the default values.

### Pathway-level information extractor (PLIER)

PLIER (*Mao et al., 2019*) is a matrix decomposition method that uses prior biological knowledge of pathways and gene sets to deconvolve gene expression profiles as a product of a small number of latent variables (factors) and their gene weights. We performed PLIER analysis using the dedicated R package (v.0.99.0; downloaded from https://github.com/wgmao/PLIER) with the collection of 5933 gene sets as a prior information matrix priorMat available in the package comprising canonical, immune and chemgen pathways from MSigDB (*Liberzon et al., 2011*), and various cell-type markers from multiple sources. The prior information matrix used as an input for this analysis is available from the PLIER analysis folder in the GitHub https://github.com/liiskolb/coexpression-transEQTL/tree/master/analysis/PLIER (*Kolberg, 2020*; copy archived at https://github.com/elifesciences-publications/coexpression-transEQTL). PLIER was run with 100 iterations (max.iter = 100). Only the 16,440

genes appearing in both gene expression data and the pathway annotation matrix were used as input. For every input gene expression matrix we analysed, the initial number of latent variables (parameter k) was set using the num.pc function provided by the PLIER package.

### Probabilistic estimation of expression residuals (PEER)

PEER (*Stegle et al., 2012*; *Stegle et al., 2010*) is a factor analysis method that uses Bayesian approaches to infer hidden factors from gene expression data that explain a large proportion of expression variability. We applied PEER method for co-expression analysis using the peer R package (v.1.0; downloaded from https://github.com/PMBio/peer) with default parameters, accounting also for the mean expression using the function PEER_setAdd_mean. The initial number of factors set by the function PEER_setNk was determined using the num.pc function from the PLIER package on each of the gene expression matrices we analysed. Therefore, the initial number of factors was set to the same value in case of PLIER and PEER. Since the number of modules estimated by the num.pc function were always larger than the number of detected modules, then this parameter value should not have a large impact on the final set of modules.

## Relationship between the eigengene profiles of co-expression clustering and matrix factorisation methods

All eigengene profiles, regardless of the co-expression analysis method used, are linear combinations of expression levels of genes belonging to the corresponding module. In case of matrix factorisation methods, each gene is assigned a weight of belonging to a module, but for a specific module a vast majority of the genes are assigned weights that are close to zero. Similarly, one could think as if the weights of genes not belonging to a specific module from co-expression clustering analysis are set to zero.

We simulated a small example to show that, in a very simplified setting, the eigengenes from matrix factorisation and co-expression clustering analysis are highly correlated. First, we generated a data matrix (300 rows and 30 columns) that includes three orthogonal vectors plus noise that represent three different gene modules of sizes 150, 100 and 50 genes. Then, we performed principal component analysis (PCA) using the R function prcomp on the standardised full matrix (each gene has unit variance) as well as on the three submatrices (mimicking three clusters detected by co-expression clustering). In the first case, we extracted the eigenvectors of the first three principal components and in the second case, we extracted only the eigenvectors of first principal components from each of the three submatrices (*Figure 1—figure supplement 6*). We found that these two approaches yielded almost identical eigenvectors (up to a scaling factor).

## Functional enrichment analysis

We used the g:GOSt tool from the g:Profiler toolset (*Raudvere et al., 2019*) via dedicated R package gprofiler2 (v.0.1.8) for functional enrichment analysis of gene modules. The short links to the full enrichment results were automatically generated using the parameter as_short_link = T in the function gost. The results shown in this paper were obtained with data version e99_eg46_p14_55317af. In case of general characterisation of gene modules and *trans* genes, we limited the data sources to Gene Ontology, Reactome and KEGG.

## *Cis*-eQTL analysis and fine mapping

We performed *cis*-eQTL analysis using the qtlmap (https://github.com/eQTL-Catalogue/qtlmap) Nextflow (*Di Tommaso et al., 2017*) workflow developed for the eQTL Catalogue project (*Kerimov et al., 2020*). Briefly, we performed *cis*-eQTL analysis in a +/- 1 Mb window centered around each gene. We used the first six principal components (PCs) of both the gene expression and genotype data as covariates in the analysis. The number of genotype PCs was chosen based on the GTEx V8 analysis, which used the first five PCs (*Aguet et al., 2019*). While the number of gene expression PCs to be included in the analysis is sometimes optimised on each dataset to maximise eQTL discovery, we have found that beyond including the first few principal components the results usually change only minimally. The eQTL analysis was performed using QTLtools (*Delaneau et al., 2017*).

For *cis*-eQTL fine mapping, we used the Sum of Single Effects (SuSiE) model (*Wang et al., 2018*) implemented in the susieR v0.9.0 R package. We performed fine mapping on a +/- 1 Mb *cis* window centered around the lead eQTL variant of each gene using individual-level genotype and gene expression data. Prior to fine mapping, we regressed out six principal components of the gene expression and genotype data from the gene expression data. To identify significant eQTLs for QTL mapping, we performed Bonferroni correction for each gene to account for the number of variants tested per gene and then used Benjamini-Hochberg FDR correction to identify genes with FDR < 0.1. The fine mapping Nextflow workflow for *cis*-eQTLs is available from GitHub (https://github.com/eQTL-Catalogue/susie-workflow).

## Gene module *trans*-eQTL analysis and fine mapping

The MatrixEQTL (*Shabalin, 2012*) R package (v2.2) was used for *trans*-eQTL analysis to fit a linear model adjusted for sex, batch (where available) and the first three principal components of the genotype data. Before the analysis, the module eigengene profiles were transformed using the inverse normal transformation to reduce the impact of outlier eigengene values produced by some clustering methods. A total of 6,861,056 autosomal genetic variants with minor allele frequency (MAF) >0.05 were tested. Due to the partial sharing of individuals between cell types and conditions, the eQTL analysis was performed in each cell type and condition separately. To achieve this, the eigenvectors from the integrated approach were split into cell-type-specific sub-eigenvectors before the analysis. The results from every analytical setting (data partitioning approach (n = 2), co-expression method (n = 5), cell type (n = 9), 90 *trans*-eQTL analyses in total) were then individually filtered to keep nominally significant variant-module associations (p-value<$5\times10^{-8}$).

Next, we applied SuSiE (*Wang et al., 2018*) to fine map the nominally significant associations to independent credible sets of variants. For every gene module, we started fine mapping from the lead variant (variant with the smallest association p-value for this module) and used a +/- 500,000 bp window around the variant to detect the credible sets. We continued fine mapping iteratively with the next best nominally significant variant outside the previous window to account for LD and continued this process until no variants remained for the gene module. This procedure resulted in a total of 864 credible sets across all cell types, co-expression analysis methods and data partitioning approaches (integrated and separate).

To aggregate and summarise overlapping associations, we combined all credible sets into an undirected graph where every node represents a credible set of a module from a triplet (data partitioning approach, co-expression method, cell type) and we defined an edge between two nodes if the corresponding credible sets shared at least one overlapping variant (*Figure 1—figure supplement 2*). The graph was constructed using the igraph R package. After obtaining the graph, we searched for connected components, that is subgraphs where every credible set is connected by a path, to combine the vast number of results into a list of non-overlapping loci (n = 601), which can no longer be interpreted as credible sets. For every component, we defined the lead variant by choosing the intersecting variant with the largest average posterior inclusion probability (PIP) value across all the credible sets in the component.

Genes in physical proximity often have correlated expressions levels and could thus manifest as co-expression modules in our analysis. Consequently, if one or more genes in such modules have *cis*-eQTLs, then these *cis* variant-module associations would also be detected by our approach. To differentiate *cis*-acting co-expressions module eQTLs from true *trans* associations, we decided to add an additional filtering step based on gene-level analysis. We performed gene-level eQTL analysis for individual gene expression traits of the 18,383 protein-coding genes and the 601 lead variants. The gene-level eQTL analysis was performed using the MatrixEQTL R package with the same settings and data transformations as in the module-level analysis described above. From every credible set component, we excluded the variant-module pairs together with corresponding credible sets where no *trans* associations (variant-level Benjamini-Hochberg FDR 5%) were included in the module. As *trans*-eQTLs we consider variants that act on distant genes (>5 Mb away from the lead variant) and genes residing on different chromosomes. Furthermore, we performed one-sided Fisher's exact tests to assess the significance of overlap between the modules and gene-level *trans* analysis associations and excluded the variant-module pairs that did not have significant overlap with individual *trans* genes (Bonferroni-adjusted p-value<0.05) (*Supplementary file 2*). After this filtering step we

repeated the process of aggregating credible sets, retaining 247 non-overlapping loci (*Figure 1—figure supplement 5*; *Supplementary file 1*).

To further account for the number of co-expression modules tested, we applied both Benjamini-Yekutieli false discovery rate (BY FDR) and Bonferroni correction at the level of each analytical setting (*Figure 1—figure supplement 5*). We applied the BY FDR 10% threshold to every module - lead variant pair from each of the 90 analytical settings (data partitioning approach, co-expression analysis method, cell type) and if a pair did not pass the threshold we excluded it together with the corresponding credible set(s) from the results. Bonferroni correction was applied in a similar manner with a threshold P-value $< \frac{5 \times 10^{-8}}{n_i}$, where $n_i$, $i = 1, ..., 90$, stands for the number of modules from the corresponding co-expression method and data partitioning approach. We repeated the graph-based aggregation process on the remaining credible sets individually from both correction methods and as a result the BY FDR 10% threshold reduced the number of significant associations to 38 and Bonferroni threshold to only 3 significant *trans*-eQTLs.

## Colocalisation

We downloaded GWAS summary statistics for 36 blood cell traits (*Astle et al., 2016*) from the NHGRI-EBI GWAS Catalog (*Buniello et al., 2019*). We downloaded coloc (*Giambartolomei et al., 2014*) R package v3.1 from bioconda (*Grüning et al., 2018*). The *cis*-eQTL colocalisation Nextflow workflow is available from GitHub (https://github.com/kauralasoo/colocWrapper). The same workflow was adjusted for *trans*-eQTL colocalisation.

## *cis*-eQTL summary statistics for *SLC39A8*

(*Kim-Hellmuth et al., 2017*) profiled gene expression in monocytes before and after stimulation with LPS, muramyl-dipeptide (MDP) and 5′-triphosphate RNA for 90 min and 6 hr. We downloaded the *cis*-eQTL summary statistics from ArrayExpress (*Athar et al., 2019*) (accession E-MTAB-5631). Individual-level genotype data were not available for this study.

## Mediation analysis

We performed the mediation analysis using the R package mediation (v. 4.5.0) (*Tingley et al., 2014*). In case of *ARHGEF3*, *LYZ* and *SLC39A8,* we performed mediation tests for trios (*trans*-eQTL variant, *cis* gene, *trans* module) using a non-parametric bootstrap method (1000 simulations) for p-value and 95% confidence interval estimations of effects implemented in the package. We used the *cis*-gene expression as mediator, the *trans*-eQTL variant as the exposure variable and the module eigenvector as the outcome variable. We also included the same covariates as in the *trans*-eQTL mapping to the models. The analysis results include estimates for average causal mediation effects (ACME, also known as indirect effects), average direct effects (ADE) and for total effects (ACME + ADE). We considered that the *cis* gene partially mediates the gene module *trans* association if the estimates of indirect effects were statistically significant.

## Replication of genetic associations

For the identified *trans*-eQTLs (after filtering for BY FDR <10%), we compared associated modules in unstimulated monocytes, neutrophils and T-cells to matched cell types from three independent studies: BLUEPRINT (*Chen et al., 2016*), ImmVar (*Raj et al., 2014*) and *Quach et al., 2016*. The genotype and expression data from these studies were accessed and processed as described previously (*Kerimov et al., 2020*).

We performed gene-level *trans*-eQTL analysis in these data for only the associations we detected in matching cell types after filtering for BY FDR <10%, that is, 9 out of 38 lead variants. The analysis was performed using the MatrixEQTL R package with the same settings and data transformations as in the module-level analysis described above. For comparison with the modules, we used the list of significant genes (variant-level Benjamini-Hochberg FDR 5%) to perform one-sided Fisher's exact test to estimate the significance of the overlap.

For literature-based replication of *LYZ*, *IFNB1* and *ARHGEF3*, we extracted the corresponding *trans* genes from the independent studies where available (see *Table 1*). For each locus we evaluated the pairwise overlap between the associated modules and previously reported genes using one-sided Fisher's exact test. If not provided by the study, we used g:Convert tool from g:Profiler

(*Raudvere et al., 2019*) to map the *trans* genes to unique Ensembl IDs before the comparisons. For every locus we adjusted the p-values using Bonferroni correction across the modules.

We compared the modules associated with the *ARHGEF3* locus with the genes reported in eQTL-Gen database (*Võsa et al., 2018*) and by *Nath et al., 2017*. The eQTLGen Consortium (*Võsa et al., 2018*) performed *trans*-eQTL analysis for 10,317 trait-associated genetic variants in 31,684 whole blood samples. For *Figure 2—figure supplement 2*, we downloaded the summary statistics from https://www.eqtlgen.org/trans-eqtls.html. We extracted the results of *Nath et al., 2017* from their supplementary material. Similarly, the results for comparing modules associate with *LYZ* and *IFNB1* loci from *Quach et al., 2016*, *Rotival et al., 2011* and *Rakitsch and Stegle, 2016* were extracted from corresponding supplementary materials provided by the studies.

## Acknowledgements

We thank Urmo Võsa, Silva Kasela, Kaido Lepik and Sina Rüeger for helpful comments on the manuscript. The computational analyses were performed at the High Performance Computing Center, University of Tartu.

## Additional information

### Funding

| Funder | Grant reference number | Author |
| --- | --- | --- |
| Eesti Teadusagentuur | PSG59 | Liis Kolberg<br>Hedi Peterson |
| Eesti Teadusagentuur | MOBJD67 | Kaur Alasoo |
| Horizon 2020 Framework Programme | 825775 | Kaur Alasoo |
| Eesti Teadusagentuur | IUT34-4 | Hedi Peterson<br>Kaur Alasoo |
| Eesti Teadusagentuur | PSG415 | Kaur Alasoo |
| European Regional Development Fund | EXCITE | Liis Kolberg<br>Nurlan Kerimov<br>Hedi Peterson<br>Kaur Alasoo |

The funders had no role in study design, data collection and interpretation, or the decision to submit the work for publication.

### Author contributions

Liis Kolberg, Conceptualization, Data curation, Software, Formal analysis, Visualization, Methodology, Writing - original draft, Writing - review and editing; Nurlan Kerimov, Data curation, Software, Visualization; Hedi Peterson, Supervision, Funding acquisition; Kaur Alasoo, Conceptualization, Data curation, Software, Supervision, Funding acquisition, Methodology, Writing - original draft, Project administration, Writing - review and editing

### Author ORCIDs

Liis Kolberg https://orcid.org/0000-0002-0118-7562
Kaur Alasoo https://orcid.org/0000-0002-1761-8881

### Ethics

Human subjects: Gene expression and genotype data from the CEDAR study were available for download without restrictions from ArrayExpress. For the Fairax_2012, Fairfax_2014 and Naranbhai_2015 studies we applied for access via the relevant Data Access Committee. For the Kasela_2017, we obtained approval from the Data Access Committee of the Estonian Biobank. Ethical approval for the project was obtained from the Research Ethics Committee of the University of Tartu (approval 287/T-14).

**Decision letter and Author response**
Decision letter https://doi.org/10.7554/eLife.58705.sa1
Author response https://doi.org/10.7554/eLife.58705.sa2

# Additional files
## Supplementary files
• Supplementary file 1. Co-expression *trans*-eQTL analysis results together with links to g:Profiler enrichment analysis (XLSX).

• Supplementary file 2. Module credible sets and overlaps with gene-level trans-eQTL analysis results (XLSX).

• Supplementary file 3. Functional enrichment evaluation of associated gene modules and gene-level analysis results (XLSX).

• Supplementary file 4. Mediation analysis results for LYZ, ARHGEF3 and SLC39A8 (XLSX).

• Supplementary file 5. Literature replication results (XLSX).

• Transparent reporting form

## Data availability
The gene expression and genotype data from the CEDAR study have been deposited to ArrayExpress under accession codes E-MTAB-6666 and E-MTAB-6667. The gene expression data from the Kasela_2017 study have been deposited to GEO under the accession code GSE78840, the individual level genotype data can be accessed upon ethical approval by submitting a data release request to the Estonian Genome Center, University of Tartu. The gene expression data from the Fairfax_2012, Fairfax_2014 and Naranbhai_2015 studies have been deposited to ArrayExpress under accession codes E-MTAB-945, E-MTAB-2232 and E-MTAB-3536. The corresponding genotype data have been deposited to EGA under accession codes EGAD00010000144 and EGAD00010000520. The gene expression matrix, detected gene modules, eigenvectors and trans-eQTL credible sets are available in Zenodo (https://doi.org/10.5281/zenodo.3759693).

The following dataset was generated:

| Author(s) | Year | Dataset title | Dataset URL | Database and Identifier |
|---|---|---|---|---|
| Kolberg L, Alasoo K | 2020 | Co-expression trans-eQTL analysis data | https://doi.org/10.5281/zenodo.3759693 | Zenodo, 10.5281/zenodo.3759693 |

The following previously published datasets were used:

| Author(s) | Year | Dataset title | Dataset URL | Database and Identifier |
|---|---|---|---|---|
| Dimitrieva J, Georges M | 2018 | Genotyping of 323 healthy Europeans blood samples (CEDAR cohort) for cis-eQTL analysis in 6 immune cell types and ileal, colonic and rectal biopsies | https://www.ebi.ac.uk/arrayexpress/experiments/E-MTAB-6666/ | ArrayExpress, E-MTAB-6666 |
| Dimitrieva J, Georges M | 2018 | Identification of cis-eQTLs in six immune cell types CD4, CD8, CD14,CD15, CD19, PLA and ileal, colonic and rectal biopsies in 323 healthy European individuals (CEDAR cohort) | https://www.ebi.ac.uk/arrayexpress/experiments/E-MTAB-6667/ | ArrayExpress, E-MTAB-6667 |
| Milani L, Peterson P | 2017 | Pathogenic Implications for Autoimmune Mechanisms Derived by Comparative eQTL Analysis of CD4+ Versus CD8+ T cells | https://www.ncbi.nlm.nih.gov/geo/query/acc.cgi?acc=GSE78840 | NCBI Gene Expression Omnibus, GSE78840 |
| Fairfax BP, Knight JC | 2012 | Genetics of gene expression in primary immune cells | https://www.ebi.ac.uk/arrayexpress/experiments/E-MTAB-945/ | ArrayExpress, E-MTAB-945 |

| Fairfax BP, Knight JC | 2014 | Genetics of gene expression across innate immune stimulation in primary monocytes | https://www.ebi.ac.uk/arrayexpress/experiments/E-MTAB-2232/ | ArrayExpress, E-MTAB-2232 |
| Fairfax BP, Knight JC, Naranbhai V | 2015 | Genomic modulators of gene expression in human neutrophils | https://www.ebi.ac.uk/arrayexpress/experiments/E-MTAB-3536/ | ArrayExpress, E-MTAB-3536 |
| Fairfax BP, Knight JC | 2012 | Genetics of gene expression in primary human immune cells | https://www.ebi.ac.uk/ega/datasets/EGAD00010000144 | European genome-phenome Archive, EGAD00010000144 |
| Fairfax BP, Knight JC | 2014 | Genetics of gene expression across innate immune stimulation in primary monocytes | https://www.ebi.ac.uk/ega/datasets/EGAD00010000520 | European genome-phenome Archive, EGAD00010000520 |

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
