## [Decision Letter]

**Acceptance summary:**

In this manuscript, Kolberg et al. identified *trans*-eQTLs from isolated immune cell and blood gene expression data from thousands of samples isolated from individuals of European descent. After careful quality control, they used different approaches to identify co-expressed gene modules in these datasets, and used these modules for the eQTL scans. The careful analyses, including replication across different studies, indicates that the co-expression module-level approach is reliable for discovering *trans*-eQTLs with broad effects on gene expression.

**Decision letter after peer review:**

Thank you for submitting your article "Co-expression analysis reveals interpretable gene modules controlled by *trans*-acting genetic variants" for consideration by *eLife*. Your article has been reviewed by three peer reviewers, including Stephen CJ Parker as the Reviewing Editor and Reviewer #1, and the evaluation has been overseen by Patricia Wittkopp as the Senior Editor. The following individual involved in review of your submission has agreed to reveal their identity: Helene Ruffieux (Reviewer #3).

The reviewers have discussed the reviews with one another and the Reviewing Editor has drafted this decision to help you prepare a revised submission.

As the editors have judged that your manuscript is of interest, but as described below that additional analyses are required before it is published, we would like to draw your attention to changes in our revision policy that we have made in response to COVID-19 (https://elifesciences.org/articles/57162). First, because many researchers have temporarily lost access to the labs, we will give authors as much time as they need to submit revised manuscripts. We are also offering, if you choose, to post the manuscript to bioRxiv (if it is not already there) along with this decision letter and a formal designation that the manuscript is "in revision at *eLife*". Please let us know if you would like to pursue this option. (If your work is more suitable for medRxiv, you will need to post the preprint yourself, as the mechanisms for us to do so are still in development.)

Summary:

In this manuscript, Kolberg et al. identified *trans*-eQTLs from isolated immune cell and blood gene expression data from thousands of samples isolated from ~1000 individuals of European descent. They assessed the expression levels of ~18,000 genes. After careful quality control, they used five different approaches to identify co-expressed gene modules in these datasets. They identified nearly 4,000 co-expression modules. The sizes of these modules ranged from tens of genes to hundreds. They then summarized the expression of module genes by essentially calculating the first principle component of the module gene expression and labeled this summary statistic as the eigengene. They calculated the association of the eigengene with 6.8 million variants using linear regression and identified ~600 loci significant at the GWAS nominal threshold of 5e-8. They were able to determine that strong *cis* eQTLs were responsible for the associations of small modules. When they eliminated these *cis*-eQTL driven modules, they were left with ~300 associations. Correcting for multiple testing with an FDR procedure or Bonferroni approach reduced the number of significant associations to 140 and 4, respectively. They reassuringly identified two known *trans* eQTLs (*IFNB1* and *LYZ*). The authors then focus on the *ARHGEF3* locus, which was identified in a large blood eQTL meta-analysis, but here the authors show that the signal is platelet specific. This is an interesting approach to identify *trans* eQTL hotspots. All reviewers uniformly felt this work is of interest, but is in need of improvement in several areas. We list these below. If these areas can be satisfactorily addressed, the collection of candidate hits should constitute a valuable resource for generating hypotheses on eQTL regulation in specific cell types, to be explored in further research.

Essential revisions:

1) For the *trans* eQTL loci identified with the co-expression approach, the authors need to map the expression of individual genes (which they do) and then assess the overlap between the genes in the modules and the genes that significantly map to the same loci individually. The significance of association for each of the genes would not necessarily be genome-wide significant (say 5-e8) but the authors can relax the significance criteria at various p-value thresholds and assess the overlap between the module genes and individually mapped genes. If there is a significant overlap, this further strengthens their argument that eigengene mapping is a useful approach to detect additional *trans* eQTLs that cannot be detected with individual gene mapping.

2) The pros and cons of the different co-expression methods should be commented more extensively, in light of the data and question asked. The authors should discuss how the specificities of each method are reflected in the uncovered modules; the fact that conclusions are obtained from multiple methods does not justify eluding this discussion. For instance, Figure 1D seems to indicate that WGCNA tends to estimate fewer modules compared to funExplorer, any explanation why? Moreover, most of the co-expression methods involve a large number of tuning parameters. Although these parameters are provided in the Methods section, the strategy for choosing them is not described (data-driven? pilot analyses? are the default parameters always used, and if so, is it justified? etc) and the extent to which this may impact inference is not discussed. Finally, how do the different types of "eigengenes" produced by the co-expression methods (factor loadings, PCs, etc) affect eQTL mapping?

3) The authors mention: "In addition to replicating a number of established *trans*-eQTL loci". This is vague, can replication rates be provided? Given that *trans* associations are particularly difficult to uncover, this information would be particularly useful to assess the potential of the approach. The current discussion focuses on dissecting the two loci *ARHGEF3* and *SLC39A8* and does not allow one to fully appreciate the overall effectiveness of the proposed module-based eQTL mapping. Replication rates for the uncovered hits may be easily obtained: e.g., using the independent study from Kim-Hellmuth et al. for monocytes (which the authors use to confirm signals for the *SLC39A8* locus) and, for the cell types with expression measured in two independent datasets (Table 1), one could "discover" the effects in the first dataset and "validate" them in the second dataset.

4) The authors rightly point out that module-based eQTL mapping reduces multiple testing. However, given that a same gene can contribute to multiple modules and that several co-expression methods are used on the same data, another complex source of multiplicity is introduced which would also require proper adjustment. This has not been addressed nor acknowledged. At the very least, a caveat should be formulated.

5) Another layer of complexity arises from the parallel analysis of datasets for each cell type and an "integrated" dataset combining all cell types. Hence, the same samples are analysed twice and the eQTL significance thresholds used in the paper again do not correct for this.

6) What is the overlap between the genes that map to the *IFNB1* and *LYZ* loci in previous *trans* eQTL studies and the genes in the co-expression modules whose eigengenes mapped to these loci in this study? Is there a significant overlap? This should be reported. If there is no significant overlap, the potential reasons for this should be discussed.

7) What is the overlap among the gene membership of the three modules whose eigengene mapped to the *ARHGEF3* locus? This is mentioned in the text but it is not clear how many genes are in each of the three modules. This locus was previously identified in a blood eQTL analysis. What is the overlap with the genes identified in the blood study and this study? If there is no significant overlap, the potential reasons for this should be discussed.

8) The authors should perform a conditional analysis, such as causal inference modeling, network edge orienting, mendelian randomization, etc to identify if the *cis*-associated gene really regulates the *trans*-associated gene expression.

---

## [Author Response]

Essential revisions:1) For the trans eQTL loci identified with the co-expression approach, the authors need to map the expression of individual genes (which they do) and then assess the overlap between the genes in the modules and the genes that significantly map to the same loci individually. The significance of association for each of the genes would not necessarily be genome-wide significant (say 5-e8) but the authors can relax the significance criteria at various p-value thresholds and assess the overlap between the module genes and individually mapped genes. If there is a significant overlap, this further strengthens their argument that eigengene mapping is a useful approach to detect additional trans eQTLs that cannot be detected with individual gene mapping.

This is an excellent suggestion. Indeed, our original criteria of requiring only one gene to overlap between the co-expression module and gene-level analysis did not properly account for the fact that some modules are quite large and therefore a very small overlap might be purely due to chance. We now assessed the significance of overlap between the gene-level *trans* genes (BH FDR 5%) and every module associated with the corresponding locus (303 loci in total) using a one-sided Fisher’s exact test.

Applying a more stringent threshold for each locus by requiring the associated modules to significantly overlap (Bonferroni adjusted Fisher’s exact p-value < 0.05) with the gene-level *trans* analysis results reduces the number of independent loci from 303 to 247. We replaced the previous filter (at least one *trans* gene in a module) with this more stringent threshold also in the manuscript. We included a Supplementary file 2 that for every locus-module pair includes the statistics of overlap sizes with the gene-level analysis.

2) The pros and cons of the different co-expression methods should be commented more extensively, in light of the data and question asked. The authors should discuss how the specificities of each method are reflected in the uncovered modules; the fact that conclusions are obtained from multiple methods does not justify eluding this discussion. For instance, Figure 1D seems to indicate that WGCNA tends to estimate fewer modules compared to funExplorer, any explanation why?

Based on this comment we included a section to the Results comparing the modules detected by different methods in more detail:

“The number of detected modules and their sizes varied due to the properties and the default parameters of each method (Figure 1D). […] Similarly, funcExplorer detected more modules than WGCNA (Figure 1D) probably because funcExplorer was able to detect modules containing fewer genes (minimum of 5 versus 20 genes) if these were supported by functional enrichment (Figure 1——figure supplement 1).”

Moreover, most of the co-expression methods involve a large number of tuning parameters. Although these parameters are provided in the Materials and methods section, the strategy for choosing them is not described (data-driven? pilot analyses? are the default parameters always used, and if so, is it justified? etc) and the extent to which this may impact inference is not discussed.

We have now modified the subsection ‘Co-expression analysis’ in the Materials and methods in multiple places to clarify which parameters were used and what was the strategy for choosing the parameters. Specifically, we now also state that:

“To reduce complexity, we relied on default parameters recommended by the authors of each co-expression analysis method. Exceptions to this are stated below.”

Initially, we decided to avoid extensive parameter searches as we felt that the “goodness” of identified modules could only be evaluated after completing the full *trans*-eQTL analysis and this would have been infeasible. However, after our manuscript was submitted, Way et al. [1] published a very through benchmarking multiple gene co-expression analysis methods, demonstrating that there is no single best method or set of parameters for co-expression analysis with different methods and parameter settings discovering biological features at different levels of granularity. In light of this, we have now added two references to Way et al. to the Introduction:

“[…] with recent benchmarks demonstrating that there is no single best co-expression analysis method (Way et al., 2020). Thus, applying multiple co-expression methods to the same dataset can aid *trans*-eQTL detection by identifying complementary sets of co-expression modules capturing a wider range of biological processes (Way et al., 2020).”

Finally, we agree with the reviewers that it would be interesting to try the five co-expression analysis methods that we have used here with a wide range of parameter settings to see how this affects *trans*-eQTL discovery, but we feel it would be out of the scope of the current manuscript. Moreover, if different parameter settings identify different co-expression modules and thus different *trans*-eQTLs, it would still remain unclear how to properly integrate those results and account for multiple testing (as the reviewers correctly highlight in point 4 below).

Finally, how do the different types of "eigengenes" produced by the co-expression methods (factor loadings, PCs, etc) affect eQTL mapping?

One of the challenges of working with multiple co-expression analysis methods is that they often use very different terms to describe similar concepts, which makes it harder to realise how the methods are related to each other. This useful comment from the reviewers helped us realise that there is actually a straightforward connection between the factor loadings used by matrix factorisation methods (PLIER, ICA and PEER) and the first principal component (eigenvector) of the co-expression module used by the co-expression clustering methods (WGCNA and funcExplorer).

Let us consider a standardised gene expression matrix where each gene has zero mean and unit variance. For simplicity, let us assume that this matrix contains three orthogonal gene modules each containing a different number of genes. Now, if we perform a principal component analysis (simplest matrix factorisation approach) of this matrix, then the loadings of the first principal component (PC) will correspond to the eigengene of the largest co-expression module, the loadings of the second PC will correspond to the eigengene of the second-largest co-expression module and the loadings of the third PC will correspond to the eigengene of the third and final co-expression module. Similarly, if we cluster this gene expression matrix into three clusters corresponding to the three co-expression modules, then now the first PC of each cluster (and co-expression module) will correspond to its eigengene. To illustrate this behaviour, we now describe a small simulation experiment in Materials and methods (subsection “Relationship between the eigengene profiles of co-expression clustering and matrix factorisation methods”).

Thus, even though matrix factorisation and co-expression clustering seem to be using different eigengene definitions, they are actually calculating a very similar property.

3) The authors mention: "In addition to replicating a number of established trans-eQTL loci". This is vague, can replication rates be provided? Given that trans associations are particularly difficult to uncover, this information would be particularly useful to assess the potential of the approach. The current discussion focuses on dissecting the two loci ARHGEF3 and SLC39A8 and does not allow one to fully appreciate the overall effectiveness of the proposed module-based eQTL mapping. Replication rates for the uncovered hits may be easily obtained: e.g., using the independent study from Kim-Hellmuth et al. for monocytes (which the authors use to confirm signals for the SLC39A8 locus) and, for the cell types with expression measured in two independent datasets (Table 1), one could "discover" the effects in the first dataset and "validate" them in the second dataset.

We have now clarified the in the Abstract that we replicate three established *trans*-eQTL loci (*LYZ*, *IFNB1* and *ARHGEF3*). To substantiate this claim, we have performed additional analysis and added the following new section to the Results:

“Replication of associations in independent datasets

We first performed a literature-based replication to measure the overlap between the modules that map to the loci near *IFNB1*, *LYZ* and *ARHGEF3* with the genes reported by previous studies (Table 1, Supplementary file 5). […] Only one module, IC68, did not have a significant overlap but this could be due to its large size and the fuzzy definition of the ICA module membership. For *ARHGEF3* we also compared the modules with the 163 *trans* genes reported by (Nath et al., 2017) where only the module X6.WIERENGA_STAT5A_TARGETS_DN had a significant overlap.”

Unfortunately, we were not able to use Kim-Hellmuth et al. to replicate our *trans* associations, because neither the *trans*-eQTL summary statistics nor individual-level genotype data were available from that study. However, we were able to perform additional replication analysis for nine loci detected in neutrophils, T-cells and naive monocytes using three independent datasets (ImmVar, BLUEPRINT and Quach et al) for which we had access to individual-level genotype data. This new analysis is now also described in the Results (see also response to point 3):

“To further assess the replication of identified *trans*-eQTLs (after filtering with BY FDR 10%), we compared associated modules in unstimulated monocytes, neutrophils and T-cells to matched cell types from three independent studies for which we had access to individual-level data: BLUEPRINT (Chen et al., 2016), ImmVar (Raj et al., 2014) and Quach et al. (Quach et al., 2016). […] Since this *trans*-eQTL was previously detected by Rotival et al., this suggests that in addition to small sample sizes of the replication studies, there might be biological differences in how the samples were collected.”

Finally, we decided not to split our dataset by study and perform replication in that way, because this would have allowed us to replicate only half of associations that were detected by separate approach as the co-expression analysis in the integrated approach crucially depended on jointly analysing the full gene expression matrix. Moreover, reducing our discovery sample size in half would have probably allowed us to only assess replication for the strongest associations at the *IFNB1* and *LYZ* loci that we already have evidence for from the literature.

4) The authors rightly point out that module-based eQTL mapping reduces multiple testing. However, given that a same gene can contribute to multiple modules and that several co-expression methods are used on the same data, another complex source of multiplicity is introduced which would also require proper adjustment. This has not been addressed nor acknowledged. At the very least, a caveat should be formulated.

We agree with this comment and we have added the following caveat to the Discussion:

“To maximise module discovery, we aggregated results from five co-expression analysis methods and two partitions of the same underlying (integrated versus separate). […] In our subsequent follow-up analyses, we only focused on four loci that we could either replicate in independent datasets (*IFNB1*, *LYZ*, *ARHGEF3*) or find significant support from the literature (*SLC39A8*).”

5) Another layer of complexity arises from the parallel analysis of datasets for each cell type and an "integrated" dataset combining all cell types. Hence, the same samples are analysed twice and the eQTL significance thresholds used in the paper again do not correct for this.

See response to point 4.

6) What is the overlap between the genes that map to the IFNB1 and LYZ loci in previous trans eQTL studies and the genes in the co-expression modules whose eigengenes mapped to these loci in this study? Is there a significant overlap? This should be reported. If there is no significant overlap, the potential reasons for this should be discussed.

We included a new section “Replication of associations in independent datasets” to the manuscript summarising the overlaps with previous *trans*-eQTL studies (where available) in case of *ARHGEF3*, *IFNB1* and *LYZ* loci. We measured the pairwise overlap between the corresponding associated modules and the *trans* genes detected by previous studies. We evaluated the significance of the overlaps using one-sided Fisher’s exact test and applied threshold of Bonferroni adjusted P-value < 0.05. If Ensembl IDs were not provided by the study, we used g:Convert tool to map the *trans* genes to unique Ensembl IDs before comparing with our gene modules. We added the following paragraph to the Results:

“We first performed a literature-based replication to measure the overlap between the modules that map to the loci near *IFNB1*, *LYZ* and *ARHGEF3* with the genes reported by previous studies (Table 1, Supplementary file 5). […] For *ARHGEF3* we also compared the modules with the 163 *trans* genes reported by (Nath et al., 2017) where only the module X6.WIERENGA_STAT5A_TARGETS_DN had a significant overlap.”

7) What is the overlap among the gene membership of the three modules whose eigengene mapped to the ARHGEF3 locus? This is mentioned in the text but it is not clear how many genes are in each of the three modules. This locus was previously identified in a blood eQTL analysis. What is the overlap with the genes identified in the blood study and this study? If there is no significant overlap, the potential reasons for this should be discussed.

See response to point 6 for more details. The sizes of the three modules are described in the first paragraph of the Section “Platelet specific *trans*-eQTL at the *ARHGEF3* locus is associated with multiple platelet traits” and the sizes of the overlaps with previously published gene lists are shown in Supplementary file 5.

8) The authors should perform a conditional analysis, such as causal inference modeling, network edge orienting, mendelian randomization, etc to identify if the cis-associated gene really regulates the trans-associated gene expression.

Our colocalisation analysis suggests that for loci near *ARHGEF3*, *LYZ* and *SLC39A8* the *cis* genes might play a role in the regulation of corresponding *trans* associated gene modules (see also Figure 2A, Figure 1—figure supplement 4A, Figure 3A). We performed mediation analysis using the R package mediation (v 4.5.0) to further assess this. That is, we performed mediation tests for trios (SNP, *cis* gene, *trans* module) using a non-parametric bootstrap method (1,000 simulations) implemented in the package for P-value and confidence interval estimation. We also included the same covariates we used in eQTL mapping.

We have added the following paragraph to the Results:

“For three of the four *trans*-eQTL loci discussed above (*LYZ*, *ARHGEF3* and *SLC39A8*), we also detected an overlapping *cis*-eQTL effect on one or more *cis* genes. […] Finally, multiple independent causal variants in the region that are in LD with each other could bias the mediation estimates (Figure 3A).”

To evaluate the mediation of *LYZ*, we performed mediation analysis for gene modules associated with the lead variant rs10784774 (chr12_69344099_A_G) in unstimulated monocytes and in monocytes after 24 h of stimulation with IFNγ. In total, we analysed 57 mediation trios with *LYZ* gene expression as mediator and found 8 modules that showed significant mediation effect (Bonferroni corrected P-value < 0.05), 7 of which were detected in IFNγ stimulated monocytes (Supplementary file 4).

In case of *IFNB1* we did not observe significant colocalising *cis* association in our analysis and therefore the assumptions for mediation analysis do not hold.